# Machine Learning-Based Agoraphilic Navigation Algorithm for Use in Dynamic Environments with a Moving Goal

Hasitha Hewawasam, Gayan Kahandawa *  and Yousef Ibrahim

School of Engineering, Information Technology and Physical Sciences, Federation University Australia, Churchill, VIC 3842, Australia

* Correspondence: g.appuhamillage@federation.edu.au

**Abstract:** This paper presents a novel development of a new machine learning-based control system for the Agoraphilic (free-space attraction) concept of navigating robots in unknown dynamic environments with a moving goal. Furthermore, this paper presents a new methodology to generate training and testing datasets to develop a machine learning-based module to improve the performances of Agoraphilic algorithms. The new algorithm presented in this paper utilises the free-space attraction (Agoraphilic) concept to safely navigate a mobile robot in a dynamically cluttered environment with a moving goal. The algorithm uses tracking and prediction strategies to estimate the position and velocity vectors of detected moving obstacles and the goal. This predictive methodology enables the algorithm to identify and incorporate potential future growing free-space passages towards the moving goal. This is supported by the new machine learning-based controller designed specifically to efficiently account for the high uncertainties inherent in the robot's operational environment with a moving goal at a reduced computational cost. This paper also includes comparative and experimental results to demonstrate the improvements of the algorithm after introducing the machine learning technique. The presented experiments demonstrated the success of the algorithm in navigating robots in dynamic environments with the challenge of a moving goal.

**Keywords:** agoraphilic; free-space; mobile robots; navigation; dynamic environment; moving goal; machine learning; fuzzy logic



## 1. Introduction

Mobile robot navigation plays a vital role in the field of robotics. The Agoraphilic navigation algorithm has been developed to overcome many challenges in mobile robot navigation [1–3]. The work presented in this paper is an advanced development of Agoraphilic algorithm.

There are a number of path planning methods developed for robot navigation [4–7]. Among them, artificial potential field (APF) [8], cell decomposition [9], mathematical programming [10] and roadmap [11] are identified as fundamental path planning algorithms. The APF method is popular among researchers due to its many advantages, such as simplicity and adaptability. However, there are well-documented inherited problems in APF-based methods [7,12,13]. These inherited drawbacks, such as (i) trap situations or dead locks (local minima), (ii) no passage between closely spaced obstacles, (iii) oscillations in narrow corridors and (iv) the goal non-reachable-with-obstacles-nearby problem (GN-RON), have motivated researchers to improve the APF-based methods and overcome these problems [14–17]. However, all these methods have attempted to address the bad outcomes of the APF method while keeping its basic concept. As a result, those algorithms have lost some of the main advantages of the APF method such as simplicity and adaptability.

The novel Agoraphilic algorithm was developed to reduce the drawbacks of APF method while keeping its advantages. It imitates human navigation behavior to reach the goal. In contrast to APF method, the Agoraphilic algorithm does not look for obstacles to

avoid but for space 'solutions' to follow, hence the term 'Agoraphilic'. For this reason, it is termed an 'optimistic' navigation algorithm.

Only a few navigation algorithms can track and hunt a moving goal in an unknown dynamic environment [4]. However, there are practical applications where a robot has to follow or hunt a moving goal. The authors' previous work on developing the new Agoraphilic navigation algorithm in dynamic environments (ANADE) also had a limitation of tracking and hunting a moving goal. Therefore, in this research, we further improved the ANADE algorithm using a machine learning (ML)-based method to address this issue. This paper presents the novel ML-based ANADE algorithm which incorporates the free-space attraction concept to track and hunt a moving goal in an unknown dynamic environment. To successfully follow and hunt a moving goal it is essential to track and estimate the location and velocity of the moving goal. Furthermore, having a short-term path prediction of the moving goal will increase the efficiency of the task.

This paper discusses the overall development of the new algorithm while providing detailed information about the new ML-based system introduced for ANADE algorithm. Section 2 of this paper discusses the development of ML-based ANADE algorithm. Section 3 introduces the new ML-based system. Development of the experimental configuration and results are presented in Section 4. Section 5 provides a discussion of our findings and concluding remarks.

## 2. Development of the ML-Based Agoraphilic Navigation Algorithm in Dynamic Environment (ANADE)

The architecture of the ML-based ANADE is a modular-based architecture [1]. This architecture allows the modification of individual modules while keeping the main algorithm unchanged. This approach enhances the resilience of the algorithm for future applications. The new ANADE algorithm consists of nine main modules.

1.　Sensory Data Processing (SDP) module;
2.　Moving goal tracking module;
3.　Moving goal path prediction module;
4.　Obstacle tracking module;
5.　Obstacle path prediction module;
6.　Current Global Map (CGM) generation module;
7.　Future Global Map (FGM) generation module;
8.　Free-space attraction module;

　　(a)　Free-space histogram generation module;
　　(b)　Free-space force generation module;
　　(c)　Force shaping module;
　　(d)　Instantaneous driving force component (Fc) generation module;

9.　Instantaneous driving force component weighing module.

These modules are iteratively used in the proposed algorithm to navigate the robot toward a moving goal in a dynamic environment, Figure 1. Additionally, the fundamental sequence of steps in the algorithm is elaborated using pseudocode in Section 2.1.

### 2.1. Pseudocode of ML-Based ANADE with a Moving Goal

- Step 1: Begin the algorithm.
- Step 2: Collect sensory data and input it into the SDP module.
- Step 3: Input the processed sensory data into the moving goal tracking module.
- Step 4: The moving goal tracking module estimates the location and velocity of the moving goal.
- Step 5: Input the processed sensory data into the obstacle tracking module.
- Step 6: The obstacle tracking module estimates the locations and velocities of moving obstacles.

- Step 7: Using the current location of the goal, the current locations of moving obstacles and static obstacles, CGM generation module generates the Current Global Map (CGM) as shown in Figure 1.
- Step 8: Predict future locations of the goal based on its estimated current location and velocity using the moving goal path prediction module.
- Step 9: Predict future locations of the obstacles based on their estimated current locations and velocities using the obstacle path prediction module.
- Step 10: Generate a set of future global maps (FGMs) using a set of future locations of the goal, sets of future locations of moving obstacles, location of static obstacles and states of the robot, Figure 1.
  Note : After Step 9 there is one current global map and multiple future global maps. Each of these maps proceeds through Steps 11 to 14, which are the components of free-space attraction module.
- Step 11: Identify the free-space passages in the global map and convert the global map into a free-space histogram.
- Step 12: Convert the generated free-space histogram into a set of free-space forces.
- Step 13: Use the ML-based force shaping module to focus the free-space forces towards the goal.
- Step 14: Feed the set of shaped forces into the instantaneous driving force component generation module. This module produces a single force for the particular global map known as the instantaneous driving force component (Fc,n).
- Step 15: Feed all the instantaneous driving force components (Fc,1, Fc,2, ..., Fc,N) into the instantaneous driving force component weighing module to create the instantaneous driving force for the current iteration. This is the robot's actual driving force.
- Step 16: If the robot has reached the goal, go to Step 17. If not, go back to Step 2.
- Step 17: Stop the algorithm.

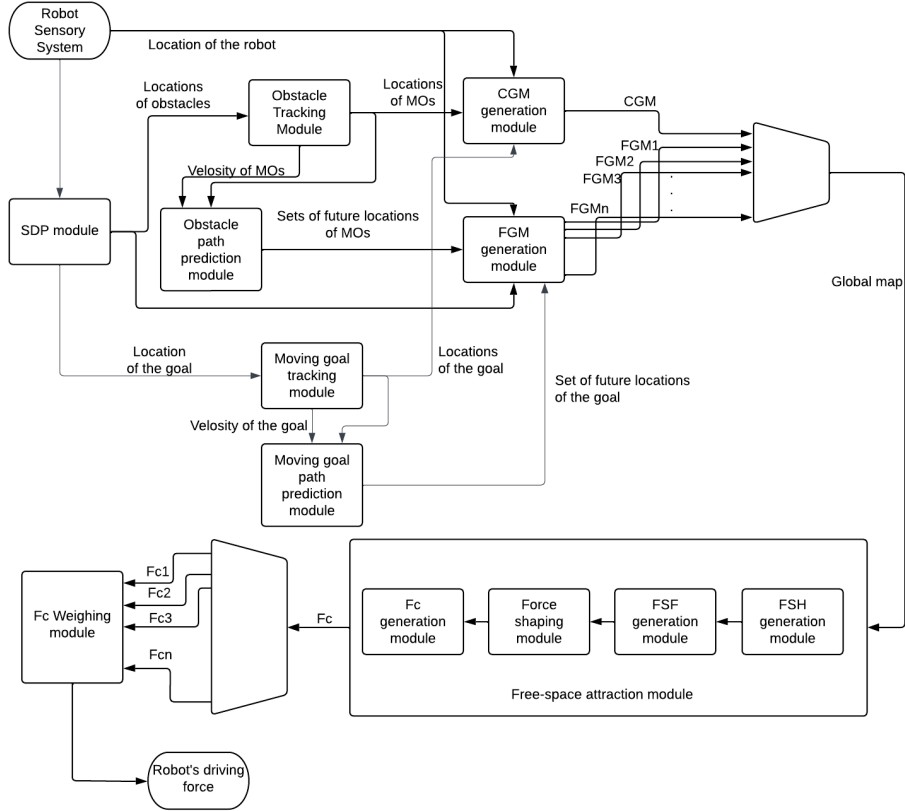

**Figure 1.** Block diagram of ML-based ANADE.

### 2.2. Sensory Data Processing (SDP) Module

The SDP module takes data from the robot's sensory system. The robot's sensory system consists of a 360° LiDAR sensor, a depth camera and an inertial measurement unit (IMU). The SDP module converts the data received from the sensory system into the world reference frame.

### 2.3. Moving Goal Tracking Module

The SDP module feeds the data into the goal tracking module. The goal tracking module tracks the moving goal and estimates the states (velocity and position) of the moving goal. This module runs a Kalman Filter (KF) -based algorithm to produce its outputs (i.e., the positions and velocity of the moving goal).

The CGM generation module and goal path prediction module take these outputs as their inputs.

In the tracking module, the kinematic model of moving goal/target is defined as follows (Equation (1)):

$$
\begin{bmatrix} x_k \\ y_k \\ \dot{x}_k \\ \dot{y}_k \end{bmatrix} = \begin{bmatrix} 1 & 0 & dt & 0 \\ 0 & 1 & 0 & dt \\ 0 & 0 & 1 & 0 \\ 0 & 0 & 0 & 1 \end{bmatrix} \begin{bmatrix} x_{k-1} \\ y_{k-1} \\ \dot{x}_{k-1} \\ \dot{y}_{k-1} \end{bmatrix} +
$$

$$
\begin{bmatrix} a_x \times \frac{dt^2}{2} \\ a_y \times \frac{dt^2}{2} \\ a_x \times dt \\ a_y \times dt \end{bmatrix} \times u(t) + w_k \tag{1}
$$

$$
\begin{bmatrix} x_k \\ y_k \end{bmatrix} = \begin{bmatrix} 1 & 0 & 0 & 0 \\ 0 & 1 & 0 & 0 \end{bmatrix} + v_k.
$$

The prior estimation (state estimation based on the previous state estimation) and globalized sensory data (measurements $(x_{k,m}, y_{k,m})$ are combined using the filter shown in Equation (2) to derive the optimal state estimations for each moving obstacle in each iteration.

$$
\begin{bmatrix} x_k \\ y_k \\ \dot{x}_k \\ \dot{y}_k \end{bmatrix} = \begin{bmatrix} 1 & 0 & dt & 0 \\ 0 & 1 & 0 & dt \\ 0 & 0 & 1 & 0 \\ 0 & 0 & 0 & 1 \end{bmatrix} \begin{bmatrix} x_{k-1} \\ y_{k-1} \\ \dot{x}_{k-1} \\ \dot{y}_{k-1} \end{bmatrix} +
$$

$$
\begin{bmatrix} a_x \times \frac{dt^2}{2} \\ a_y \times \frac{dt^2}{2} \\ a_x \times dt \\ a_y \times dt \end{bmatrix} + k_k \left\{ \begin{bmatrix} x_{k,m} \\ y_{k,m} \end{bmatrix} - \right.
$$

$$
\begin{bmatrix} 1 & 0 \\ 0 & 1 \\ 0 & 0 \\ 0 & 0 \end{bmatrix}^T \left( \begin{bmatrix} 1 & 0 & dt & 0 \\ 0 & 1 & 0 & dt \\ 0 & 0 & 1 & 0 \\ 0 & 0 & 0 & 1 \end{bmatrix} \begin{bmatrix} x_{k-1} \\ y_{k-1} \\ \dot{x}_{k-1} \\ \dot{y}_{k-1} \end{bmatrix} \right.
$$

$$
\left. \left. + \begin{bmatrix} a_x \times \frac{dt^2}{2} \\ a_y \times \frac{dt^2}{2} \\ a_x \times dt \\ a_y \times dt. \end{bmatrix} \right) \right\} \tag{2}
$$

In this expression, $k_k$ (Kalman gain) is found by using Equation (3);

$$
k_k = p_k \begin{bmatrix} 1 & 0 \\ 0 & 1 \\ 0 & 0 \\ 0 & 0 \end{bmatrix}^T \left( \begin{bmatrix} 1 & 0 \\ 0 & 1 \\ 0 & 0 \\ 0 & 0 \end{bmatrix}^T p_k \begin{bmatrix} 1 & 0 \\ 0 & 1 \\ 0 & 0 \\ 0 & 0 \end{bmatrix} \right)^{-1},
\tag{3}
$$

where:

$$
p_k = \begin{bmatrix} 1 & 0 & dt & 0 \\ 0 & 1 & 0 & dt \\ 0 & 0 & 1 & 0 \\ 0 & 0 & 0 & 1 \end{bmatrix} p_{k-1} \begin{bmatrix} 1 & 0 & 0 & 0 \\ 0 & 1 & 0 & 0 \\ dt & 0 & 1 & 0 \\ 0 & dt & 0 & 1 \end{bmatrix} + Q.
\tag{4}
$$

At each iteration following the optimal state estimation of every moving obstacle, $p_k$ is updated as shown in Equation (5).

$$
p_k = \begin{bmatrix} 1 & 0 & 0 & 0 \\ 0 & 1 & 0 & 0 \\ 0 & 0 & 1 & 0 \\ 0 & 0 & 0 & 1 \end{bmatrix} - k_k \begin{bmatrix} 1 & 0 & dt & 0 \\ 0 & 1 & 0 & dt \\ dt & 0 & 1 & 0 \\ 0 & dt & 0 & 1 \end{bmatrix} \times
$$
$$
p_{k-1} \begin{bmatrix} 1 & 0 & dt & 0 \\ 0 & 1 & 0 & dt \\ dt & 0 & 1 & 0 \\ 0 & dt & 0 & 1 \end{bmatrix} + Q.
\tag{5}
$$

The updated $p_k$ is used as $p_{(k-1)}$ in Equation (4) to re-calculate the new $p_k$ in the next iteration.

where:

$$
\begin{aligned}
v_k &\sim N(0,R) \\
w_k &\sim N(0,Q) \\
p &\overset{\text{def}}{=} \text{error covariancet} \\
(x_k, y_k) &\overset{\text{def}}{=} \text{estimated position of} \\
&\quad\text{the moving obstacle} \\
(x_{km}, y_{km}) &\overset{\text{def}}{=} \text{measured position of} \\
&\quad\text{the moving obstacle} \\
(a_x, a_y) &\overset{\text{def}}{=} \text{acceleration of the} \\
&\quad\text{moving obstacle.}
\end{aligned}
$$

### 2.4. Moving Goal Path Prediction Module

This module estimates the future locations of the goal. This allows the algorithm to revise the robot's trajectory to reach the moving goal effectively. As mentioned in the above section, current states of the moving goal (estimated by the moving goal tracking module) are taken as the main input of this module. These input data are processed by the moving goal path prediction module to create a set of future locations of the goal (this will predict up to ten future iterations). These predicted locations of the goal are used to develop the FGMs.

Future states (positions and velocities) of the moving goal are predicted using the prediction model described in Equation (6).

In this module, a simple and efficient approach is used, while a more sophisticated algorithm is used for tracking and estimating the current state of the goal. The predictions are updated at each iteration and new predictions are made according to the updated current state data. This approach allows for short-term prediction of the future path while minimizing computational complexity and maintaining low latency.

$$
\begin{bmatrix} x(t+n) \\ y(t+n) \\ dx(t+n)/dt \\ dy(t+n)/dt \end{bmatrix} = \begin{bmatrix} 1 & 0 & nT & 0 \\ 0 & 1 & 0 & nT \\ 0 & 0 & 1 & 0 \\ 0 & 0 & 0 & 1 \end{bmatrix} \begin{bmatrix} x(t) \\ y(t) \\ \frac{dx(t)}{dt} \\ \frac{dy(t)}{dt} \end{bmatrix} +
$$
$$
\begin{bmatrix} \frac{(nT)^2}{2} \times \frac{d^2x(t)}{dt^2} \\ \frac{(nT)^2}{2} \times \frac{d^2y(t)}{dt^2} \\ nT \times \frac{d^2x(t)}{dt^2} \\ nT \times \frac{d^2y(t)}{dt^2} \end{bmatrix} \tag{6}
$$

where:

$x(t) \overset{\text{def}}{=}$    position in x direction

$y(t) \overset{\text{def}}{=}$    position in y direction

$T \overset{\text{def}}{=}$    sample period

$n \overset{\text{def}}{=}$    sample number.

### 2.5. Obstacle Tracking Module

This module estimates the current velocity and position of moving obstacles. This module takes sensory data as its input and generates two outputs.

1. Positions of moving obstacles;
2. Velocity vectors of moving obstacles.

This module employs a tracking algorithm [18] developed based on Kalman filter. The obstacle path prediction module takes the two outputs of this module as its input. Further, the estimated positions of moving obstacles are fed into CGM and FGM generation modules.

### 2.6. Obstacle Path Prediction Module

This module estimates the future positions of moving obstacles. This allows the algorithms to make decisions based on future growing or diminishing free-space.

The obstacle path prediction module generates sets of predicted locations of moving obstacles. The predicted locations are fed into the FGM generation module.

### 2.7. Current Global Map (CGM) Generation Module

The CGM generation module generates a map of the robot's environment with respect to the world's axis system. This modified module takes four main inputs.

1. Locations of static obstacles with respect to the robot's axes system from the sensory system;
2. Current locations of moving obstacles from the moving obstacle tracking module;
3. Current location of the goal/target estimated from the goal tracking module;
4. Current location of the robot from the sensory system.

The CGM generation module processes these four inputs and creates a map of the environment with respect to the global axis system. The free-space attraction module takes this map (i.e., the CGM) as an input.

### 2.8. Future Global Map (FGM) Generation Module

The main task of the FGM generation module is to forecast the robot's future environment set-ups and generate a set of maps called FGMs. This module takes three main inputs,

1. The current location of the robot;
2. Sets of future positions of moving obstacles;
3. A set of future positions of the goal.

The FGM at iteration 'N' is generated by combining the estimated location of the robot at $N^{th}$ iteration with forecasted locations of moving and static obstacles and the goal at the $N^{th}$ iteration. If FGM generation module develops predicted maps till the t + n iteration, there will be n−1 FGMs. The FGMs are updated at each iteration with corrections made to position estimations.

### 2.9. Free-Space Attraction (FSA) Module

The FSA module produces a force vector out of a global map (a CGM or a FGM). This force vector is called the instantaneous driving force component ($F_c$). Four main sub-modules are used to develop the FSA modul [1,19,20].

1. Free-Space Histogram (FSH) generation module;
2. Free-Space Forces (FSF) generation module;
3. Force-shaping module;
4. Instantaneous driving force component (Fc) generation module.

Among these four sub-modules, the force shaping module has a direct influence on determining the robot's motion direction. In previous versions of ANADE algorithms, the force-shaping module was developed using either numerical or fuzzy logic-based controllers. In this paper, the previous force-shaping modules were further strengthened by the new ML-based approach. This new approach allowed the algorithm to perform efficiently with a moving goal. The development of the new force-shaping module is discussed in Section 3.

### 2.10. Instantaneous Driving Force (IDF) Generation Module

This module takes all the instantaneous driving force components as its input. The components of forces ($F_c$) are then weighted according to the accuracy of the prediction. The weighted average of instantaneous driving forces is taken as the final driving force of the current iteration.

## 3. Machine Learning (ML) for ANADE Algorithm

The force-shaping modules used in the previous versions of ANADE algorithms do not consider the velocity of the moving goal when shaping the free-space forces [1]. The novel ML-based force-shaping module was developed to address this problem. The main challenge associated with ML was generating a good training and testing datasets. The trained module with these pre-determined datasets should be able to successfully replace the fuzzy logic controller (FLC) in the force shaping module while incorporating the velocity of the moving goal for force shaping. Training and testing datasets were developed for the five main Agoraphilic modes:

1. Goal seeking: the goal is in the line of sight of the robot;
2. Normal travel: the robot is in an environment with dynamic as well as static obstacles with a reasonable amount of free-space;
3. Safe travel: the robot is in a cluttered environment with very limited free-space;
4. Right side safe: the robot is in a cluttered environment with limited free-space on the left side of the robot;
5. Left side safe: the robot is in a cluttered environment with limited free-space on the right side of the robot.

The development process of these datasets is discussed in the following sections.

### 3.1. Training and Testing Datasets Generation

The development of a novel ML-based force shaping module required the creation of a substantial dataset for training and testing purposes. However, manually creating a large dataset is challenging and impractical. To address this, five fuzzy logic controllers (FLCs) were developed to generate initial datasets, each corresponding to one of the five main Agoraphilic modes. Subsequently, domain knowledge was applied to modify the initial

datasets to obtain final training and testing datasets, Figure 2. In this stage velocity of the moving goal was considered. By making these modifications, the datasets became more representative of real-world scenarios, allowing for more accurate training and evaluation of the models across different Agoraphilic modes.

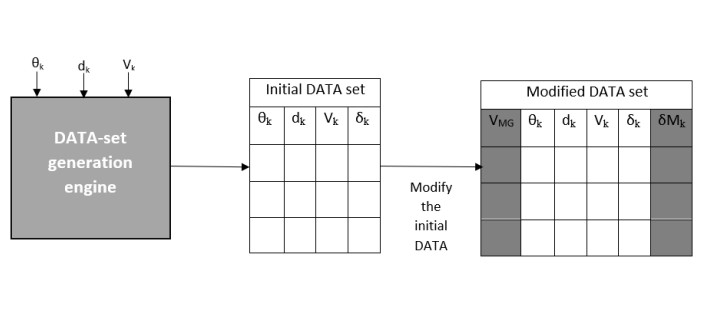

**Figure 2.** Data-set generation procedure.

### 3.2. Datasets for Different Agoraphilic Behaviors

Five dataset generation engines were developed using five different fuzzy logic controllers. Those are linguistic fuzzy logic controllers. To develop the fuzzification module, four databases were designed. Three databases were used for the three inputs:

1. angle difference between sector forces and the goal ($\theta_k$). Five different databases were used for the five engines;
2. normalized sector distance ($d_{k,n}$)-this is common for all five engines (database is developed according to Equation (12));
3. normalized sector velocity ($v_{k,n}$)-this is common for all five engines (database is developed according to Equation (13))

and one for the output,

1. fuzzy shaping factor ($\delta_k$) .

The database for $\theta_k$ consists of nine fuzzy membership functions. The corresponding linguistics are LRR: Left Rear Rear, LR: Left Rear, LS: Left Side, LF: Left Front, F: Front, RF: Right Front, RS: Right Side, RR: Right Rear and RRR: Right Rear Rear (Figure 3). As mentioned above five different databases were used for $\theta_k$ in the five dataset generation engines:

1. For goal seeking, Equation (7), Figure 3;
2. For normal travel, Equation (8), Figure 4;
3. For safe travel, Equation (9), Figure 5;
4. For right side safe, Equation (10), Figure 6;
5. For left side safe, Equation (11), Figure 7.

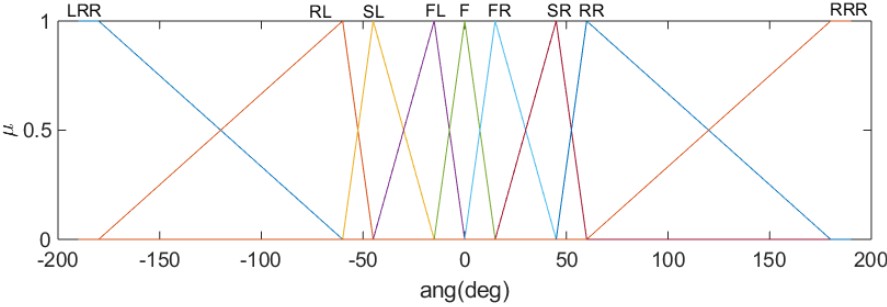

**Figure 3.** Membership function for the first input $\theta_{diff}$ of the fuzzy controller for goal seeking behavior.

$$\mu_{LRR}(\theta) = max\{min\left(\frac{-\theta - 60}{120}, 1\right), 0\}$$

$$\mu_{LR}(\theta) = max\{min\left(\frac{\theta + 180}{120}, \frac{-45 - \theta}{15}\right), 0\}$$

$$\mu_{SL}(\theta) = max\{min\left(\frac{\theta + 60}{15}, \frac{-\theta - 15}{30}\right), 0\}$$

$$\mu_{LF}(\theta) = max\{min\left(\frac{\theta + 45}{45}, \frac{-\theta}{30}\right), 0\}$$

$$\mu_{F}(\theta) = max\{min\left(\frac{\theta + 15}{15}, \frac{15 - \theta}{15}\right), 0\} \qquad (7)$$

$$\mu_{RF}(\theta) = max\{min\left(\frac{\theta}{15}, \frac{45 - \theta}{30}\right), 0\}$$

$$\mu_{SF}(\theta) = max\{min\left(\frac{\theta - 15}{15}, \frac{60 - \theta}{15}\right), 0\}$$

$$\mu_{RR}(\theta) = max\{min\left(\frac{\theta - 45}{15}, \frac{180 - \theta}{120}\right), 0\}$$

$$\mu_{RRR}(\theta) = max\{min\left(\frac{\theta - 60}{120}, 1\right), 0\},$$

where: $\theta \in \{-180, 180\}$.

$$\mu_{LRR}(\theta) = max\{min\left(\frac{-90 - \theta}{90}, 1\right), 0\}$$

$$\mu_{LR}(\theta) = max\{min\left(\frac{\theta + 180}{90}, \frac{-60 - \theta}{30}\right), 0\}$$

$$\mu_{SL}(\theta) = max\{min\left(\frac{\theta + 90}{30}, \frac{-30 - \theta}{30}\right), 0\}$$

$$\mu_{LF}(\theta) = max\{min\left(\frac{\theta + 60}{30}, \frac{-\theta}{30}\right), 0\}$$

$$\mu_{F}(\theta) = max\{min\left(\frac{\theta + 30}{30}, \frac{30 - \theta}{30}\right), 0\} \qquad (8)$$

$$\mu_{RF}(\theta) = max\{min\left(\frac{\theta}{30}, \frac{60 - \theta}{30}\right), 0\}$$

$$\mu_{SF}(\theta) = max\{min\left(\frac{\theta - 30}{30}, \frac{180 - \theta}{90}\right), 0\}$$

$$\mu_{RR}(\theta) = max\{min\left(\frac{\theta - 60}{30}, \frac{180 - \theta}{90}\right), 0\}$$

$$\mu_{RRR}(\theta) = max\{min\left(\frac{\theta - 90}{90}, 1\right), 0\},$$

where: $\theta \in \{-180, 180\}$

$$\mu_{LRR}(\theta) = max\left\{min\left(\frac{-\theta - 135}{45}, 1\right), 0\right\}$$

$$\mu_{LR}(\theta) = max\left\{min\left(\frac{\theta + 180}{45}, \frac{-90 - \theta}{45}\right), 0\right\}$$

$$\mu_{SL}(\theta) = max\left\{min\left(\frac{\theta + 135}{45}, \frac{-\theta - 45}{45}\right), 0\right\}$$

$$\mu_{LF}(\theta) = max\left\{min\left(\frac{\theta + 90}{45}, \frac{-\theta}{45}\right), 0\right\}$$

$$\mu_F(\theta) = max\left\{min\left(\frac{\theta + 45}{45}, \frac{45 - \theta}{45}\right), 0\right\} \quad (9)$$

$$\mu_{RF}(\theta) = max\left\{min\left(\frac{\theta}{45}, \frac{90 - \theta}{45}\right), 0\right\}$$

$$\mu_{SF}(\theta) = max\left\{min\left(\frac{\theta - 45}{45}, \frac{135 - \theta}{45}\right), 0\right\}$$

$$\mu_{RR}(\theta) = max\left\{min\left(\frac{\theta - 90}{45}, \frac{180 - \theta}{45}\right), 0\right\}$$

$$\mu_{RRR}(\theta) = max\left\{min\left(\frac{\theta - 135}{45}, 1\right), 0\right\},$$

where: $\theta \in \{-180, 180\}$.

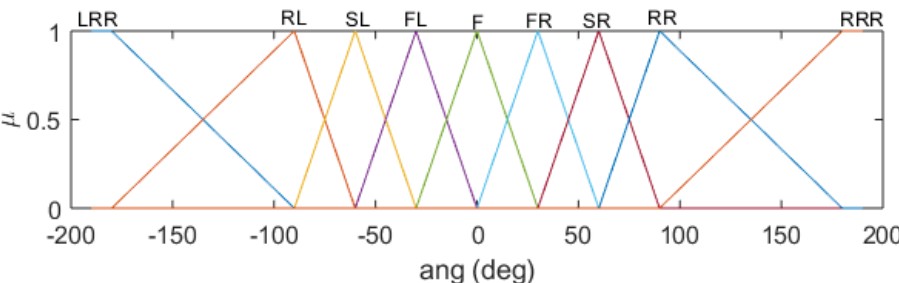

**Figure 4.** Membership function for the first input $\theta_{diff}$ of the fuzzy controller for normal travel behavior.

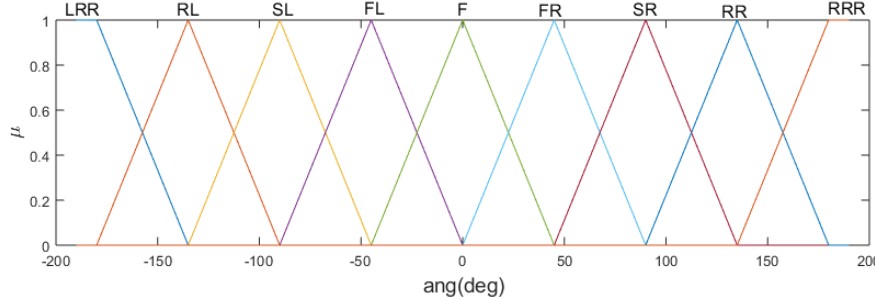

**Figure 5.** Membership function for the first input $\theta_{diff}$ of the fuzzy controller for safe travel behavior.

$$\mu_{LRR}(\theta) = max\left\{min\left(\frac{-60-\theta}{120}, 1\right), 0\right\}$$

$$\mu_{LR}(\theta) = max\left\{min\left(\frac{\theta+180}{120}, \frac{-45-\theta}{15}\right), 0\right\}$$

$$\mu_{SL}(\theta) = max\left\{min\left(\frac{\theta+60}{15}, \frac{-15-\theta}{30}\right), 0\right\}$$

$$\mu_{LF}(\theta) = max\left\{min\left(\frac{\theta+45}{45}, \frac{-\theta}{30}\right), 0\right\}$$

$$\mu_{F}(\theta) = max\left\{min\left(\frac{\theta+15}{15}, \frac{45-\theta}{45}\right), 0\right\} \qquad (10)$$

$$\mu_{RF}(\theta) = max\left\{min\left(\frac{\theta}{45}, \frac{90-\theta}{45}\right), 0\right\}$$

$$\mu_{SF}(\theta) = max\left\{min\left(\frac{\theta-45}{45}, \frac{135-\theta}{45}\right), 0\right\}$$

$$\mu_{RR}(\theta) = max\left\{min\left(\frac{\theta-90}{45}, \frac{180-\theta}{45}\right), 0\right\}$$

$$\mu_{RRR}(\theta) = max\left\{min\left(\frac{\theta-135}{45}, 1\right), 0\right\},$$

where: $\theta \in \{-180, 180\}$.

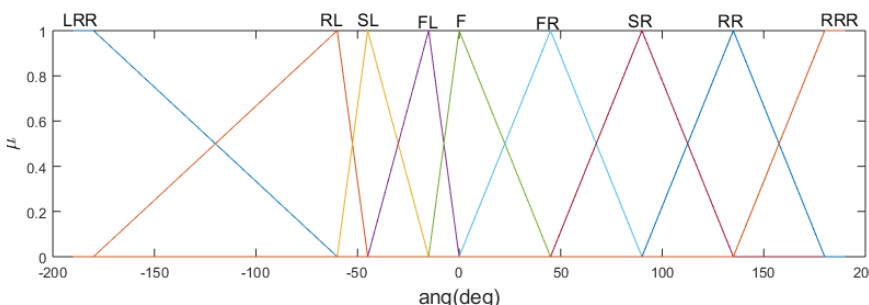

**Figure 6.** Membership function for the first input $\theta_{diff}$ of the fuzzy controller for right-side safe behavior.

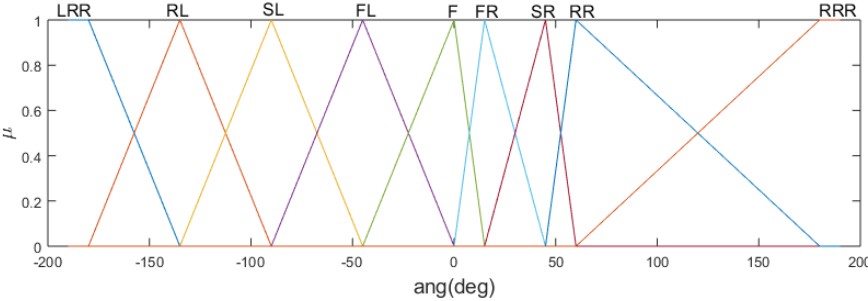

**Figure 7.** Membership function for the first input $\theta_{diff}$ of the fuzzy controller for left-side safe behavior.

$$\mu_{LRR}(\theta) = max\{min\left(\frac{-135 - \theta}{45}, 1\right), 0\}$$

$$\mu_{LR}(\theta) = max\{min\left(\frac{\theta + 180}{45}, \frac{-90 - \theta}{45}\right), 0\}$$

$$\mu_{SL}(\theta) = max\{min\left(\frac{\theta + 135}{45}, \frac{-45 - \theta}{45}\right), 0\}$$

$$\mu_{LF}(\theta) = max\{min\left(\frac{\theta + 90}{45}, \frac{-\theta}{45}\right), 0\}$$

$$\mu_{F}(\theta) = max\{min\left(\frac{\theta + 45}{45}, \frac{15 - \theta}{15}\right), 0\} \quad (11)$$

$$\mu_{RF}(\theta) = max\{min\left(\frac{\theta}{15}, \frac{45 - \theta}{30}\right), 0\}$$

$$\mu_{SF}(\theta) = max\{min\left(\frac{\theta - 15}{15}, \frac{60 - \theta}{15}\right), 0\}$$

$$\mu_{RR}(\theta) = max\{min\left(\frac{\theta - 40}{15}, \frac{180 - \theta}{120}\right), 0\}$$

$$\mu_{RRR}(\theta) = max\{min\left(\frac{\theta - 60}{120}, 1\right), 0\},$$

where: $\theta \in \{-180, 180\}$.

The database for $d_{k,n}$ also consists of five fuzzy membership functions as shown in Equation (12). The corresponding linguistics are VC: Very Close, C: Close, N: Near, F: Far, VF: Very Far, Figure 8.

$$\mu_{VC}(d_{k,n}) = max\{min\left(\frac{-d_{k,n} - 0.3}{0.2}, 1\right), 0\}$$

$$\mu_{C}(d_{k,n}) = max\{min\left(\frac{d_{k,n} - 0.1}{0.2}, \frac{0.5 - d_{k,n}}{0.2}\right), 0\}$$

$$\mu_{N}(d_{k,n}) = max\{min\left(\frac{d_{k,n} - 0.3}{0.2}, \frac{0.7 - d_{k,n}}{0.2}\right), 0\} \quad (12)$$

$$\mu_{F}(d_{k,n}) = max\{min\left(\frac{d_{k,n} - 0.5}{0.2}, \frac{0.9 - d_{k,n}}{0.2}\right), 0\}$$

$$\mu_{VF}(d_{k,n}) = max\{min\left(\frac{d_{k,n} - 0.7}{0.2}, 1\right), 0\}.$$

The database for $v_{k,n}$ also consists of seven fuzzy membership functions as shown in Equation (13). The variable $v_{k,n}$ is positive if the moving obstacle is approaching the robot. The corresponding linguistics are NF: Negative Fast, NM: Negative Medium, NS: Negative Slow, Z: Zero, PS: Positive Slow, PM: Positive Medium and PF: Positive Fast, Figure 8.

$$\mu_{NF}(v_{k,n}) = max\left\{min\left(\frac{-v_{k,n} - 0.77}{0.33}, 1\right), 0\right\}$$

$$\mu_{NM}(v_{k,n}) = max\left\{min\left(\frac{v_{k,n} + 1}{0.33}, \frac{-0.33 - v_{k,n}}{0.33}\right), 0\right\}$$

$$\mu_{NS}(v_{k,n}) = max\left\{min\left(\frac{v_{k,n} + 0.77}{0.33}, \frac{-v_{k,n}}{0.33}\right), 0\right\}$$

$$\mu_{Z}(v_{k,n}) = max\left\{min\left(\frac{v_{k,n} + 0.33}{0.33}, \frac{-0.33 - v_{k,n}}{0.33}\right), 0\right\} \tag{13}$$

$$\mu_{PS}(v_{k,n}) = max\left\{min\left(\frac{v_{k,n}}{0.33}, \frac{0.77 - v_{k,n}}{0.33}\right), 0\right\}$$

$$\mu_{PM}(v_{k,n}) = max\left\{min\left(\frac{v_{k,n} - 0.33}{0.33}, \frac{1 - v_{k,n}}{0.33}\right), 0\right\}$$

$$\mu_{PF}(v_{k,n}) = max\left\{min\left(\frac{v_{k,n} - 1}{0.33}, 1\right), 0\right\}.$$

The simple output database is represented by eight fuzzy sets, as shown in Figure 9. The fuzzy rule bases used for these dataset generation engines consist of 189 rules.

Different combinations of input data ($\theta_k$, $d_k$ and $v_k$) are fed into the five dataset generation engines and corresponding fuzzy shaping factor ($\delta_k$) were recorded in the 'initial dataset' generation process (Figure 2). This information is tabulated to get five different initial datasets for different Agoraphilic behaviors.

However, the velocity of the moving goal is not represented in those datasets. Therefore, the initial datasets were manually modified according to the domain knowledge to integrate velocity of moving goal to the dataset.

Sixty percent of the data from the new modified datasets (Figure 2) was used to train the five new ML-based modules. These modules mimic the five different Agoraphilic behaviors.

1.  ML module for goal seeking behavior;
2.  ML module for goal normal travel behavior;
3.  ML module for goal safe travel behavior;
4.  ML module for goal right side safe behavior;
5.  ML module for goal left side safe behavior.

To build the ML-based modules 'k-nearest neighbors (k-NN) algorithm' [21–23] was used.

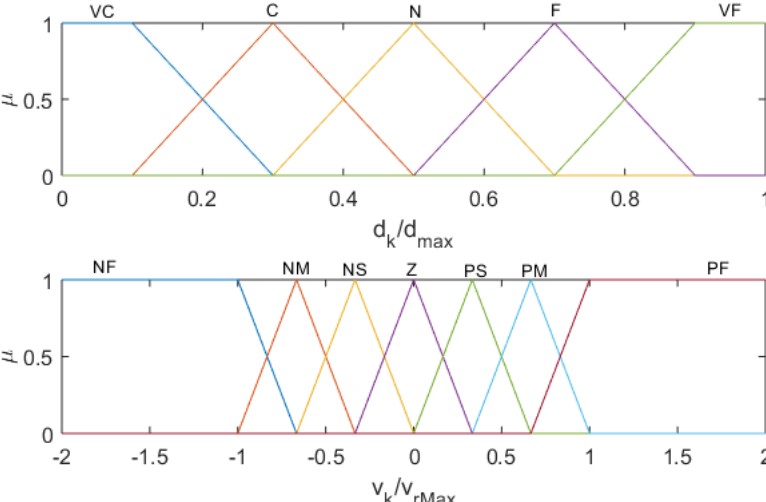

**Figure 8.** Membership functions of the second ($d_{k,n}$) and third ($v_{k,n}$) inputs of the fuzzy controller.

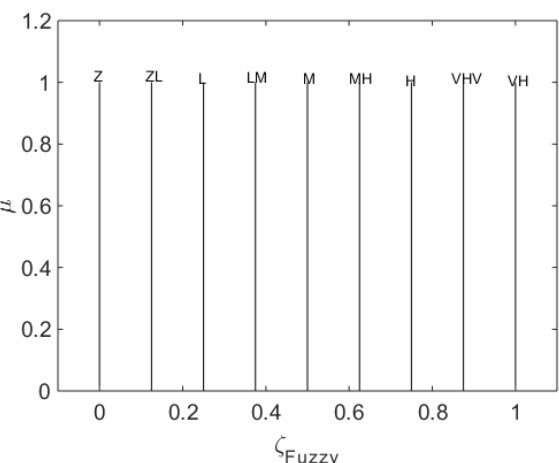

**Figure 9.** Output membership functions of the fuzzy controller.

40% data from each modified dataset was used to evaluate the corresponding trained module. In the evaluation, it was identified that the mismatch of each module was as follows (Figure 10):

1.  ML module for goal seeking behavior, 23%;
2.  ML module for normal travel behavior, 7%;
3.  ML module for safe travel behavior, 8%;
4.  ML module for right side safe behavior, 9%;
5.  ML module for left side safe behavior, 10%.

A supervisory controller alters the robot's motion behavior among the five Agoraphilic behaviors based on the different environmental conditions.

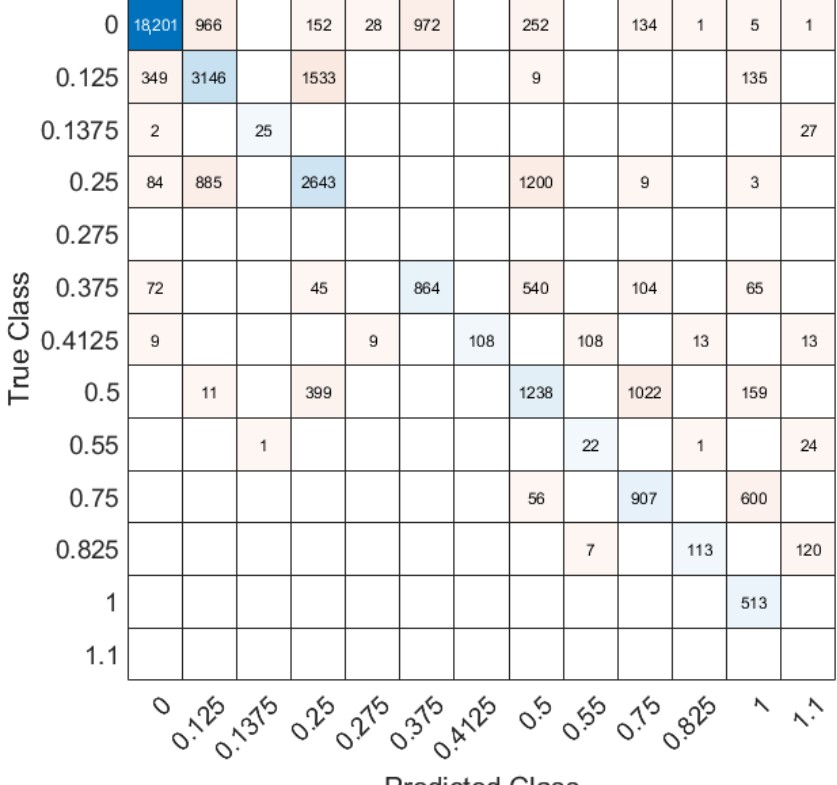

**Figure 10.** Confusion chart of ML module for goal seeking behavior.

## 4. Experimental Testing and Analysis of Results

The experimental work was conducted to demonstrate and validate the performance of the algorithm with the new ML-based force-shaping module. In this paper, a random real-world experiment is presented to validate the algorithm with the proposed force-shaping module. Furthermore, two comparative experiments are presented to test the importance of the ML-based controller. The experimental setup used is shown in Figure 11.

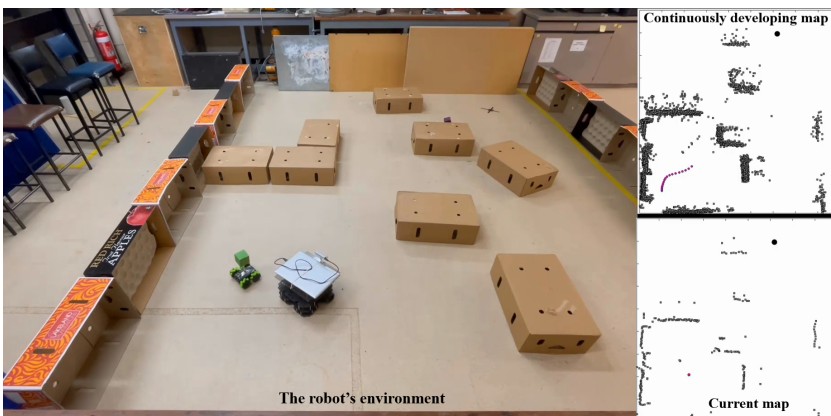

**Figure 11.** Experimental setup.

### 4.1. Experiment 1: Random Experiment with Two Moving Obstacles in a Cluttered Environment with a Moving Goal

This experiment was performed to test the algorithm's capability of reaching a moving goal in a dynamically cluttered environment with multiple moving obstacles in real-world circumstances. In this experiment, two moving obstacles with random acceleration variations were used with some static obstacles. The moving goal performed a major direction change in this experiment. The starting conditions of the experiment are shown in Table 1.

**Table 1.** Initial conditions of experiment 1.

| Object | Starting Location (cm) | Starting Velocity (cm/s) |
|---|---|---|
| Robot | (0, 0) | 0 |
| Goal | (306, −327) | $3.2\angle(141°)$ |
| Moving Obstacle 1 (MO1) | (150, 100) | $2.5\angle(−72°)$ |
| Moving Obstacle 2 (MO2) | (350, 0) | $0.81\angle(179°)$ |

The goal, the robot, and MOs started moving at the instant labelled T1, Figure 12. At this time instant, the goal was located directly in (+x, −y) direction of the robot with a linear distance of 448 cm. During T1 to T2 MO1 and MO2 developed their movements in the robot's left-hand side with the intention of challenging the robot in future iterations. At time instant T2, the robot had to change its direction because of the static obstacle in front of the robot. In this situation, the robot had 2 options,

1. Move to the left (towards MO1 and MO2);
2. Move to the right (more free-space) .

The robot picked the first option although there was more challenge from the moving obstacles. The new moving direction of the goal was the main reason for this decision. At time instant T2, the goal had changed its velocity towards +y direction. During time instants T2 to T3 the robot moved with a low speed and allowed the MO1 to pass the robot. During time instants T3 to T4 the robot moved towards the goal through the free-space passage between MO1, MO2 and the static obstacle. At time instant T4, the robot successfully reached the moving goal.

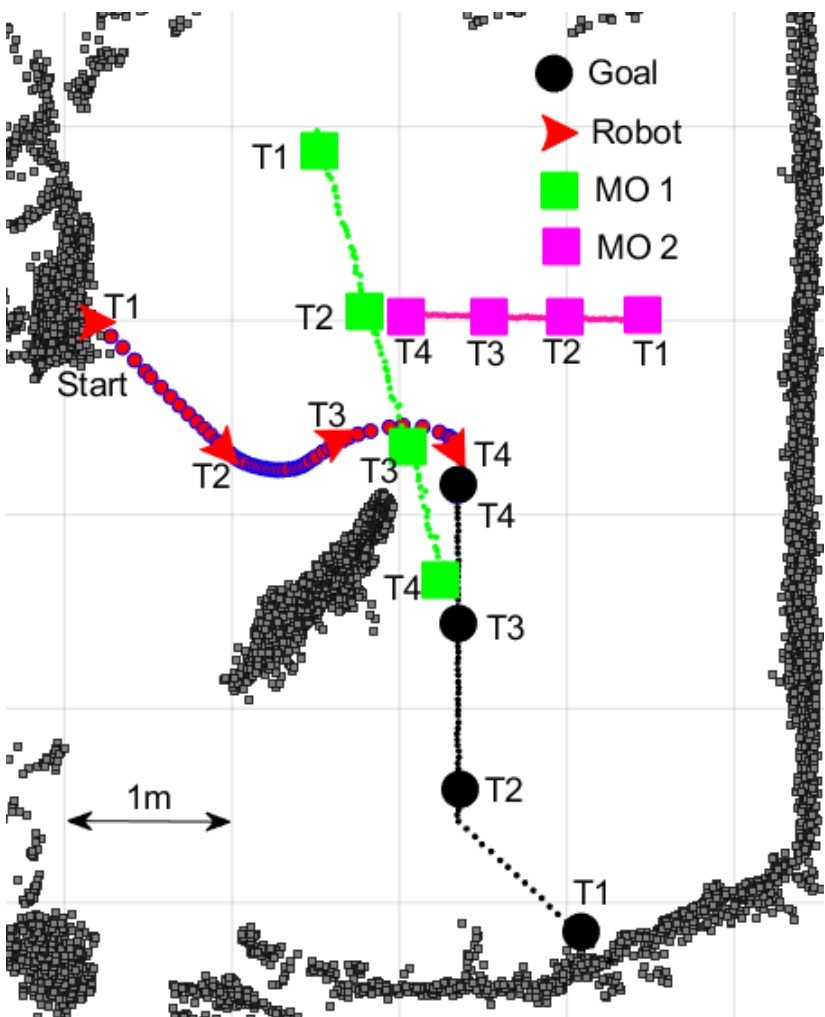

**Figure 12.** The robot's path observed in experiment 1 (robot, goal and the MOs locations in four different time instants, T1–T4).

The robot took 89 s to finish the task successfully, with a path length of 275 cm. During the process, the robot maintained an average speed of 3 cm/s. The summarized test results are shown in Table 2.

**Table 2.** Summarized test results of experiment 1.

| Parameter | Value |
|---|---|
| Robot's path length | 275 cm |
| Goal's path length | 301 cm |
| Robot's average speed | 3 cm/s |
| Goal's average speed | 3.4 cm/s |
| Total time | 89 s |

*4.2. Comparative Test between FLC-Based ANADE and ML-Based ANADE*

The experimental work presented in this subsection was conducted to demonstrate and validate the performance improvement of the ANADE algorithm with the novel ML-based force-shaping module. The experiments were designed to test the performance of the new ML-based ANADE in comparison with FLC-based ANADE. In this subsection, a simulation test and an experimental test are presented.

### 4.2.1. Comparison 1—Simulation Test

This simulation test was conducted to test the algorithm's capability of reaching a moving goal in a dynamic environment with a moving obstacle and a static obstacle. In this simulation, two tests were conducted:

1. With the ML-based force-shaping module;
2. With FLC-based force shaping module.

In both tests, all other parameters were kept unchanged. The initial conditions of the tests are shown in Table 3.

**Table 3.** Initial conditions of comparison test 1.

| Object | Starting Location (cm) | Starting Velocity (cm/s) |
|---|---|---|
| Robot | $(0, 0)$ | 0 |
| Goal | $(900, 0)$ | $2\angle(90°)$ |
| Moving Obstacle (MO) | $(150, 100)$ | $2\angle(180°)$ |

The goal, the robot, and MO started moving at the instant labelled T1, Figure 13. At this time instant, the goal was located directly in (+x, −y) direction of the robot with a distance of 900 cm. During T1 to T2, MO1 developed its movements in the robot's right-hand side with the intention of challenging the robot. At time instant T2, the robot with ML and FLC successfully passed the MO1. From T1 to T3 the robot followed the same path in both cases. The speed variations were also almost the same from T1 to T3, Figure 14. However, the robot with ML started increasing its speed as soon as it passed the static obstacle (at T3), Figure 14. Additionally, it started moving towards the moving direction of the goal. On the other hand, the robot with FLC started increasing its speed 50 iterations later. As a result of this different decision-making, the robot with ML reached the goal with a shorter path (10%), and also in less time (13.78%). Furthermore, the robot with ML had a 4.3% higher average speed. The summarized test results are shown in Table 4.

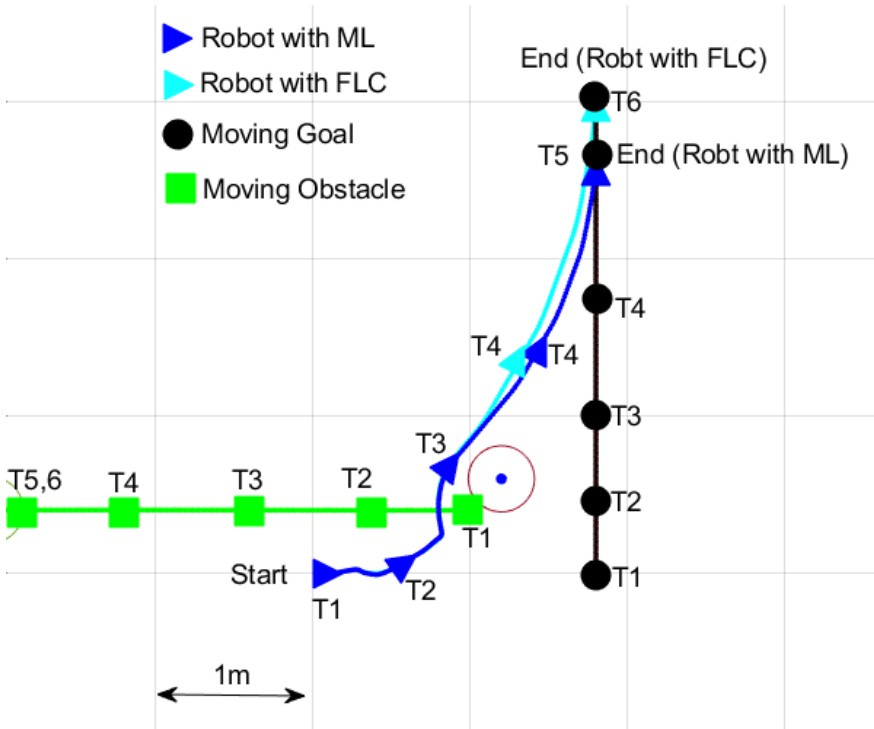

**Figure 13.** The robot's paths observed in simulation (robot, goal, and the MO locations in six different time instants, T1–T6).

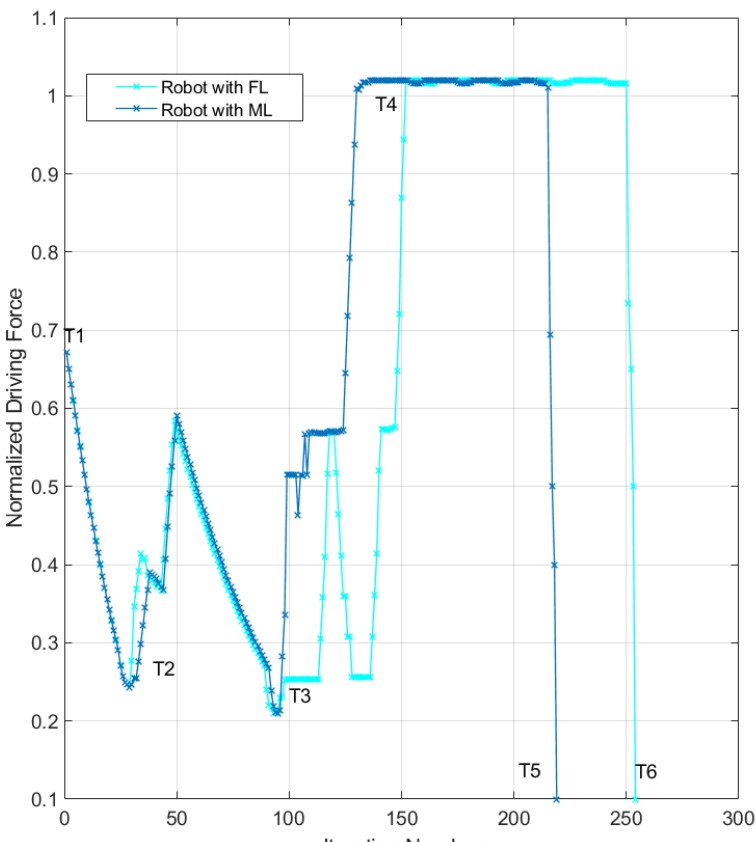

**Figure 14.** The robot's normalized driving force variations observed in simulation.

**Table 4.** Summarized test results of comparison test 1.

| Parameter | Without ML | With FLC | Performance Comparison with Respect to Case 2 |
|---|---|---|---|
| Robot's path length | 1749 cm | 1943 cm | 10% |
| Goal's path length | 1303 cm | 1512 cm | 13.78% |
| Robot's average speed | 2.68 cm/s | 2.57 cm/s | 4.3% |
| Goal's of the goal | 2 cm/s | 2 cm/s | N/A |
| Total time | 651.8 s | 756 s | 13.78% |

4.2.2. Comparison 2—Experimental Test

This experimental test was conducted to test the algorithm's capability of reaching a moving goal in a dynamic environment with multiple moving obstacles with random acceleration variations and static obstacles in the real world. Similar to comparison 1, in this experiment also two tests were conducted:

1. With the ML-based force-shaping module;
2. With FLC-based force shaping module.

In both tests, all other parameters were kept the same. The initial conditions of the experiments are tabulated in Table 5.

**Table 5.** Initial conditions of comparison test 2 (experimental test).

| Object | Starting Location (cm) | Starting Velocity (cm/s) |
|---|---|---|
| Robot | (0, 0) | $0\angle(0°)$ |
| Goal | (0, −250) | $2\angle(0°)$ |
| Moving Obstacle 1 (MO1) | (150, −100) | $1.50\angle(177°)$ |
| Moving Obstacle 2 (MO2) | (150, −100) | $1.58\angle(179°)$ |

At the time instant labelled T1, the robot, MO, and the goal started moving, Figure 15. At time instant T1, the goal was positioned towards the −y direction of the robot with a distance of 250 cm. MO1 developed its movements from T1 to T2, with the purpose of challenging the robot. At time instant T2, the robot with ML and FLC successfully passed the MO1. From T1 to T3 both the robot followed slightly different paths. The speed variations show that both the algorithms (with ML and with FLC) reduced speed as the challenge from MO2, Figure 16. The robot with FLC started changing its direction to avoid MO2. On the other hand, the robot with ML without changing its path, allowed the MO2 to pass the robot. The robot with ML increased its speed as soon as MO2 passed the robot (at T3), Figure 16. Additionally, it started moving toward the moving direction of the goal. On the other hand, the robot with FLC started increasing its speed 35 iterations later. As a result of this different decision-making, the robot with ML reached the goal with a shorter path (19%) and also in less time (35%). However, the robot with FLC had an 8% higher average speed. The summarized experimental results are tabulated in Table 6.

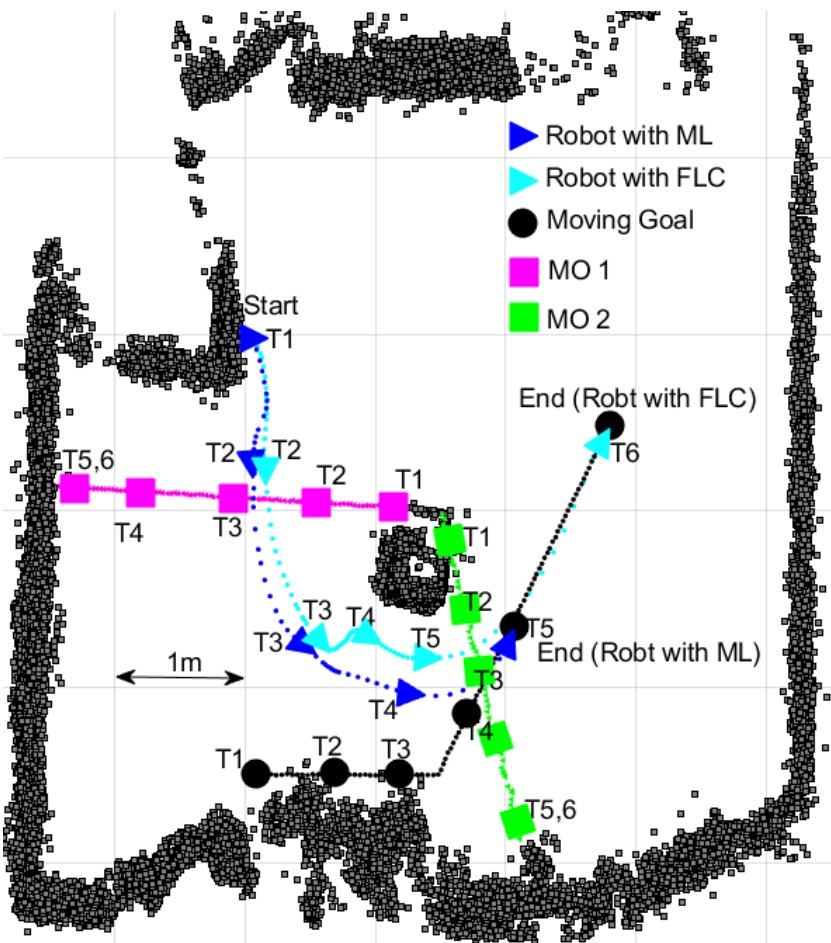

**Figure 15.** The robot's paths observed in the experimental test (robots, goal, and the MO's locations in six different time instants, T1–T4).

**Table 6.** Summarized test results of the comparison test 2 (experimental test).

| Parameter | Without ML | With FLC | Performance Comparison with Respect to Case 2 |
|---|---|---|---|
| Robot's path length | 376 cm | 467 cm | 19% |
| Goal's path length | 252 cm | 379 cm | 33% |
| Robot's average speed | 2.1 cm/s | 2.3 cm/s | 8% (low) |
| Goal's of the goal | 1.54 cm/s | 1.52 cm/s | N/A |
| Total time | 132 s | 202 s | 35% |
| Avg. sample time | 1.9 s | 2.0 s | 5% |

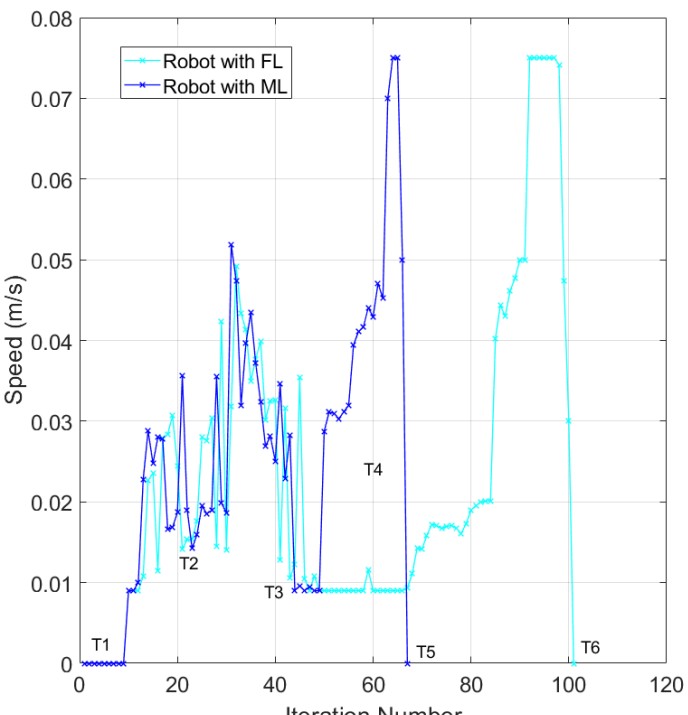

**Figure 16.** The robot's speed variations observed in simulation.

## 5. Conclusions

This paper presented the development of a novel machine learning-based Agoraphilic navigation algorithm. The new algorithm has proven its success and advantages in navigating robots in an unknown static and dynamic environment with a moving goal. The proposed algorithm uses a single attractive force to pull the robot towards the moving goal via current free-space leading to future growing free-space passages. This paper introduced a machine learning-based module to the Agoraphilic concept. This permitted the algorithm to effectively integrate the velocity of the moving goal in decision-making. To support the new machine learning-based module, the presented goal tracking, goal path prediction, CGM generation, and FGM generation modules were developed. Combining these novel modules with other main modules used in previous ANADE algorithms permits the algorithm to incorporate the rapid changes that occur in unknown dynamic environments with high uncertainties.

In this paper, a real-world experiment was presented to demonstrate the significance of the algorithm in navigating a robot with a moving goal. This experiment proved that the algorithm can successfully follow a moving target even in a dynamically cluttered environment with multiple moving obstacles. Furthermore, to demonstrate the importance of the ML technique, two comparison experiments were conducted. The experiments' results proved that the novel ML-based method improved the algorithm's performances

by reducing path length, reducing the time to hunt the moving goal and also reducing the sample time. The first comparison experiment showed that the algorithm with ML used a shorter path to reach the goal (10%), also in 13.8% less time compared to the FL-based method. Moreover, the robot with ML maintained a higher average speed (4.3%) compared to the algorithm with FL. These facts were further verified by the results observed in the second comparison experiment. The algorithm with ML reached the goal with a 19% shorter path in 35% less time. Additionally, the algorithm with ML had a 5% lower average sampling time. These comparative and experimental results demonstrated the improvements of the algorithm after introducing the machine learning technique. The presented algorithm demonstrated the enhanced capability to successfully navigate mobile robots in dynamically cluttered environments to hunt a moving target.

**Author Contributions:** H.H., Y.I. and G.K. conceived the idea. H.H. designed the methods and performed the experiments. H.H., Y.I. and G.K. analyzed the results and wrote this paper. All authors have read and agreed to the published version of the manuscript.

**Funding:** This research was supported by Australian Government Research Training Program (RTP) via Federation University of Australia.

**Institutional Review Board Statement:** Not applicable.

**Informed Consent Statement:** Not applicable.

**Data Availability Statement:** Not applicable.

**Conflicts of Interest:** The authors declare no conflict of interest.

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
