# Peer review of "Machine Learning-Based Agoraphilic Navigation Algorithm for Use in Dynamic Environments with a Moving Goal"

_machines, doi:10.3390/machines11050513_

Round 1

Reviewer 1 Report

In this paper the development of a new machine learning-based control system for the Agoraphilic (free-space attraction) concept of navigating robots in unknown dynamic environments with a moving goal is presented. A new methodology to generate training and testing datasets to develop a machine learning-based module to improve the performances of Agoraphilic algorithms is also introduced. In general, I think that the algorithmic contribution has a little bit lack on novelty; however, if I assume that the main contribution is the methodology instead of algorithmic formulation in this context this manuscript could provide a robust basis for future research and in this scenario Here some issues which I encourage the authors to consider:

1. Regarding to Section 2, the authors can consider to add a general block diagram since a high level it could be useful to the reader to understand at high level the proposed approach.

2. idem pseudocode for the main steps and how those can be implemented can be a nice complement if the current manuscript. Thus, future works could use the manuscript as basis of different variants/solutions of the path planning, obstacle avoidance and so on in the future state of the art.

3. For Section 3, it is difficult to me understand how a fuzzy logic systems can be “trained” to my knowledge fuzzy systems depends on the knowledge of the expert. Is the “training” is only for generate some test cases it is difficult to guarantee the repeatability by using only a few test cases, maybe statistical analysis of the data can complement the results, and more discussion about the fuzzy systems and how it is trained can clarify the remaining issues.

4.There are some minor typos. I recommend a detailed English revision before the manuscript can be considered for publication.

Author Response

Dear Sir/Mam,

Thank you for giving us the opportunity to submit a revised draft of the manuscript “Machine Learning-based Agoraphilic Navigation Algorithm for use in Dynamic Environments with a Moving Goal” in the Journal of “Machines”. We appreciate the time and effort that you and the reviewers dedicated to providing feedback on our manuscript and are grateful for the insightful comments on and valuable improvements to our paper. We have incorporated all the suggestions made by the reviewers. Those changes are highlighted within the manuscript. Please see attachment for a point-by-point response to reviewers’ comments and concerns.

Reviewer 2 Report

This article describes a machine learning-based control system for guiding mobile robots towards moving goals in unknown dynamic environments. The system utilizes the Agoraphilic (free-space attraction) concept to safely navigate mobile robots through cluttered environments. The algorithm estimates the position and velocity vectors of detected moving obstacles and goals using tracking and prediction strategies. It then identifies and incorporates potential growing free-space passages towards the moving goal.

1、  What are the advantages of the predictive model described in Eq. 6 over other forecasting models?

2、  The novel Agoraphilic algorithm was developed to reduce the drawbacks of APF method while keeping its simplicity. Do ML-based algorithms increase its complexity?

3、  Eq.6 appears in the four-row one-column matrix multiplied by the four-row one-column matrix, did you forget to add the transpose symbol?

4、  What steps are needed to transform the results received by the dataset generation engine into the final training and testing datasets?

5、  During the experimental verification, is the trajectory of the dynamic obstacle too simple?

Author Response

Dear Sir/Mam,

Thank you for giving us the opportunity to submit a revised draft of the manuscript “Machine Learning-based Agoraphilic Navigation Algorithm for use in Dynamic Environments with a Moving Goal” in the Journal of “Machines”. We appreciate the time and effort that you and the reviewers dedicated to providing feedback on our manuscript and are grateful for the insightful comments on and valuable improvements to our paper. We have incorporated all the suggestions made by the reviewers. Those changes are highlighted within the manuscript. Please see the attachment, for a point-by-point response to reviewers’ comments and concerns.
